# Characterization of bZIP Transcription Factors in Transcriptome of *Chrysanthemum mongolicum* and Roles of *CmbZIP9* in Drought Stress Resistance

**DOI:** 10.3390/plants13152064

**Published:** 2024-07-26

**Authors:** Xuan Wang, Yuan Meng, Shaowei Zhang, Zihan Wang, Kaimei Zhang, Tingting Gao, Yueping Ma

**Affiliations:** 1College of Life and Health Sciences, Northeastern University, Shenyang 110169, China; wangxuan99000@163.com (X.W.); 2201455@stu.neu.edu.cn (Y.M.); 2201483@stu.neu.edu.cn (S.Z.); 2301526@stu.neu.edu.cn (Z.W.); gaotingting@mail.neu.edu.cn (T.G.); 2Co-Innovation Center for Sustainable Forestry in Southern China, College of Life Sciences, Nanjing Forestry University, Nanjing 210037, China; kaimeizhang@njfu.edu.cn

**Keywords:** chrysanthemum, bZIP, transcriptome, drought tolerance, transgenic tobacco

## Abstract

bZIP transcription factors play important roles in regulating plant development and stress responses. Although bZIPs have been identified in many plant species, there is little information on the bZIPs in *Chrysanthemum*. In this study, bZIP TFs were identified from the leaf transcriptome of *C. mongolicum*, a plant naturally tolerant to drought. A total of 28 full-length bZIP family members were identified from the leaf transcriptome of *C. mongolicum* and were divided into five subfamilies based on their phylogenetic relationships with the bZIPs from Arabidopsis. Ten conserved motifs were detected among the bZIP proteins of *C. mongolicum*. Subcellular localization assays revealed that most of the CmbZIPs were predicted to be localized in the nucleus. A novel bZIP gene, designated as *CmbZIP9*, was cloned based on a sequence of the data of the *C. mongolicum* transcriptome and was overexpressed in tobacco. The results indicated that the overexpression of *CmbZIP9* reduced the malondialdehyde (MDA) content and increased the peroxidase (POD) and superoxide dismutase (SOD) activities as well as the expression levels of stress-related genes under drought stress, thus enhancing the drought tolerance of transgenic tobacco lines. These results provide a theoretical basis for further exploring the functions of the bZIP family genes and lay a foundation for stress resistance improvement in chrysanthemums in the future.

## 1. Introduction

The basic leucine zipper (bZIP) family is one of the largest families of transcription factors in plants and plays important roles in plant development and responses to various biotic and abiotic stresses. The bZIP protein is also a well-known abscisic acid (ABA)-induced DNA-binding protein family that is essential for the activation of downstream gene expression during abiotic stress [1,2,3,4,5]. This family contains a highly conserved basic domain and a relatively divergent leucine chain that bind to a core sequence (ACGT) in the promoter of the target genes, such as G-box, C-box, and A-box [4,6]. To date, many bZIP TFs have been identified in plants. For example, 869 bZIPs with nine subgroups were identified from12 species of Rosaceae [7]; 75 bZIPs with 13 subfamilies were detected in *Arabidopsis thaliana* [8]; and 62 bZIP members with 12 subgroups were found in *Andrographis paniculatan* [9]. Several bZIP members are involved in plant tolerance to salt stress, drought, etc. [10,11,12,13]. 

Chrysanthemums is a famous traditional flower in China with high ornamental, medical, and economic value. However, the production and commercialization of chrysanthemum have been hindered by water shortages [14,15]. Therefore, cultivating new chrysanthemum varieties with high drought resistance is a key goal for chrysanthemum breeders. Compared to conventional breeding approaches, genetic transformation techniques are more powerful and efficient. Although many bZIP TFs have been identified in different plants, few bZIPs have been characterized in *Chrysanthemum*. *Chrysanthemum mongolicum* belongs to *Chrysanthemum* (Asteraceae), mainly distributed in Nei Mongol, China, and is a remarkable drought-tolerant plant that grows on rocky mountain slopes at an altitude of 1500–2500 m [16,17,18]. Its outstanding drought-resistant gene resources are an excellent choice for molecular breeding for chrysanthemum stress resistance. In this study, bZIP family-member genes were identified from the leaf transcriptome of *C. mongolicum*. The protein properties, phylogenies, and motif composition were assessed. In addition, a candidate *bZIP* gene, *CmbZIP9,* was isolated from *C. mongolicum* using RT-PCR and genomic PCR. The roles of *CmbZIP* were determined through heterologous expression in *Nicotiana tabacum* L. under drought conditions. This study provides useful information on exploring the functions of the bZIP family genes in *Chrysanthemum* species and lays a foundation for stress resistance improvement in chrysanthemums in the future. 

## 2. Results

### 2.1. Phylogenetic Analysis of bZIP Family Members in C. mongolicum 

In total, 28 full-length bZIP members were identified from the transcriptome of *C. mongolicum* (Appendix A). The proteins of the bZIP family members in *C. mongolicum* range from 140 aa to 1822 aa in length. Most CmbZIPs were predicted to be localized in the nucleus (Appendix A). The bZIP family members in *C. mongolicum* were divided into five subgroups by referencing the 13 subclasses of AtbZIPs in the phylogenetic tree. The homologous proteins in families B, D, F, H, J, K, and E of *A. thaliana* were not found in the obtained transcriptome data from *C. mongolicum* (Figure 1). The largest subgroup, S, contained 10 CmbZIP members, followed by group A (8). Only one member was homologous to family G of the AtbZIPs. In addition, the homologs of the family members of groups D and E in Arabidopsis were found in other Chrysanthemum bZIPs, suggesting that some bZIPs are expressed in other organs and developmental stages. This result suggested that the members of bZIP proteins in different subgroups might be involved in various biological functions in *C. mongolicum*. 

### 2.2. Conserved Motif Analysis of CmbZIPs

A total of 10 protein-conserved motifs with 15~50 amino acid residues were identified in the CmbZIPs with low E-values (<10^−10^) using the MEME tool (Appendix A). At least one protein-conserved motif was found in all of the identified protein sequences (Figure 2). The number and type of conserved motifs in each of the five subclades were similar, but each subgroup had special characteristics. Motif 1, a conserved basic bZIP domain, was present in all of the members of the protein sequences, which confirmed the correctness of our identifications. Groups S and C had similar motif compositions, with most of the sequences containing motifs 1 and 2. We found that the protein sequences in group S had two different motif compositions: one consisted of motifs 1 and 9, and the other contained motifs 1, 2, and 8. In addition, groups A and I contained a significantly higher number of motifs than the other groups. Motif 7 predominantly existed in group I, while motifs 3 and 4 appeared only in group A. The specific motifs in each subgroup suggested their specific biological functions within the subgroup and the functional diversity of bZIP family genes.

### 2.3. Gene Isolation and Sequence Analysis

The phylogenetic analysis showed that the sequence of cluster-19700.13551 in the members of subgroup C was more closely related to that of AtbZIP9 (Figure 1), a protein that responds to drought stress. As such, we selected it to study its function in transgenic tobacco under drought stress. The full-length coding sequence of the *bZIP9* homolog, *CmbZIP9,* was successfully isolated from the leaves of *C. mongolicum* using RT-PCR and genomic PCR with a pair of primers (Appendix A). It was 768 bp in length, encoding 255 amino acids (Appendix A). The putative peptide sequence of CmbZIP9 contains a typical basic region leucine zipper (BRLZ) domain, which is conserved in bZIP family proteins (GenBank accession number OR831111, named *CmbZIP9*). *CmbZIP9* contains five exons and four introns (GenBank accession number OR842904, Appendix A). The comparison of the putative amino acid sequences of CmbZIP9 with those of other bZIP homologs revealed that the sequence identity ranged from 57.8% to 94.12%, and the highest identity of 94.12% was shared with AabZIP from *Artemisia annua* (Appendix A). The phylogenetic analysis showed that CmbZIP9 clusters with the bZIP9 homolog of *A. annua* and *Tanacetum cinerariifolium* into a clade with an 87% bootstrap support value, which is consistent with the biological evolutionary patterns (Appendix A).

### 2.4. Heterologous Expression of CmbZIP9 in Tobacco

After rooting on MS medium containing kanamycin and rifampicin, more than 65 transgenic tobacco lines were obtained. Genomic PCR and qRT-PCR were performed to confirm the *CmbZIP9* transgenic lines (Figure 3). Lines 2, 4, and 6, which had high expression levels of *CmbZIP9*, were selected for further functional studies.

### 2.5. CmbZIP9 Improved the Tolerance of Plants to Drought Stress

Four-week-old transgenic tobacco lines and wild-type seedlings were used to observe the phenotypes under drought stress. There were no obvious morphological differences between transgenic and wild-type plants under normal conditions. The WT tobacco wilted earlier than the transgenic tobacco, whereas the transgenic lines showed less wilting and grew well after 7 days of drought treatment (Figure 4A). Under drought stress, the relative water contents (RWCs) of the transgenic tobacco plants and wild-type plants were all reduced, but the transgenic tobacco plants had a significantly higher RWC than the wild-type plants (Figure 4B). 

The malondialdehyde (MDA) content gradually increased in all plants with increasing duration of drought stress, but a significantly higher content was observed in the wild-type plants than in the transgenic plants. After 7 days of exposure to drought stress, the MDA content of the wild-type plants was 1.65, 1.38 and 1.35 times higher than that in transgenic plants 2, 4 and 6, respectively (Figure 4C). Therefore, the overexpression of *CmbZIP9* reduced the degree of lipid peroxidation.

The peroxidase (POD) activity of the transgenic tobacco plants was slightly higher than that of the wild-type plants under normal conditions, after drought treatment for 4 days, and after drought treatment for 7 days. After 4 days of drought treatment, the POD contents of the transgenic tobacco plants 2, 4, and 6 was 1.18, 1.45, and 1.54 times higher than that in the wild-type plants, respectively. After 7 days of drought treatment, the POD content of transgenic plants 2, 4, and 6 was 2.33, 2.51 and 2.90 times higher than that of the wild-type plants, respectively (Figure 4D). 

Under normal conditions, the activity of superoxide dismutase (SOD) in the leaves of the transgenic plants was slightly lower than that in the wild-type plants. After drought treatment, the SOD activity of the transgenic plants increased obviously. After 4 days of drought treatment, the SOD content of the transgenic plants 2, 4, and 6 was 1.44, 1.71 and 1.61 times higher than that of the wild-type plants, respectively. After 7 days of drought treatment, the SOD content of the transgenic plants was 1.35, 1.81, and 1.91 times higher than that of the wild-type plants, respectively (Figure 4E). These results indicated that the overexpression of *CmbZIP9* in tobacco increased the tolerance of tobacco to drought stress.

### 2.6. CmbZIP9 Overexpression in Transgenic Tobacco Enhanced the Expression of Drought-Responsive Genes 

To further explore the possible molecular mechanism by which *CmbZIP9* improves drought tolerance, the transcript levels of four abiotic-stress-related genes, including ABA-responsive element binding factor (*AREB*), ethylene responsive factor *(ERF)*, S-adenosylmethionine decarboxylase (*SAMDC*), and arginine decarboxylase (*ADC*) were investigated. The results showed that the expressions of the genes related to ROS detoxification (*NtSOD*) and stress (*NtERF*, *NtAREB*, *NtADC,* and *NtSAMDC*) in the transgenic lines were significantly higher than those in the WT plants after exposure to drought stress for 7 days (Figure 5). Therefore, *CmbZIP9* might regulate the expression of stress-related genes, thus improving the tolerance of transgenic tobacco to drought stress.

## 3. Discussion

bZIP TFs have been shown to play important roles in plant development and the responses to various biotic and abiotic stress [15,19,20]. The bZIP domain is crucial to the functioning of the bZIP TF family members [21]. In this study, 28 CmbZIPs were identified in *C. mongolicum*, which were classified into five subgroups. The conserved motif of the basic region was present in all the members of the CmbZIPs, which indicated the evolutionary conservation of bZIP gene function in these plants. In addition, the CmbZIP proteins of the same subgroup shared the same conserved motif types and similar numbers of them, but the number of conserved motifs varied considerably between subclades, consistent with the bZIP findings in other plants [22,23]. The diversity of the conserved motifs outside of the bZIP domain in different bZIP members suggests that the bZIP genes might have diverse functions. 

The bZIP structure varies among species, and even among different bZIP family members of the same species. For example, a maximum of 12 introns have been detected in Arabidopsis, cucumber, and rice; 11 bZIPs were found in mung bean and barley, and so on [24,25,26,27]. In our study, four introns were found within the ORF of *CmbZIP9*. The diverse structures of *bZIPs* suggest that different splicing states of exons and introns may have been of great significance in the evolution of *bZIP* genes in plants. The positions and phases of the introns in the basic and hinge regions of the bZIP domain are highly conserved, indicating multiple functions of this gene family in plant development and biotic/abiotic stress responses. 

Usually, an appropriate balance between ROS production and scavenging is maintained in plants under normal conditions. However, large amounts of ROS are produced when plants are exposed to stress [28,29,30]. Excessive ROS results in plant metabolism disorders and oxidative damage, which inhibit normal plant growth and development and may even lead to plant death in severe cases [31,32,33]. In order to effectively reduce the accumulation of ROS, plants have evolved complex and precise antioxidant systems to neutralize ROS and prevent the oxidative damage of cells during stress conditions [34]. The MDA content is an important indicator of lipid peroxidation, which indirectly reflects the degree of plant membrane system damage and stress resistance [35,36]. The MDA content sharply increases in plants under abiotic stress [37]. In our study, the MDA content of *CmbZIP9* transgenic tobacco was significantly lower than that of the wild-type tobacco under drought stress, indicating lower cell membrane system damage and lipid peroxidation levels in *CmbZIP9* transgenic plants under drought stress than in wild-type plants.

SOD and POD are important components of the antioxidant enzyme defense system that can effectively remove excess ROS from plants [37,38,39]. SOD scavenges ROS by converting superoxide into H_2_O_2_ [40], whereas POD can consume ROS [41]. As a result, higher SOD and POD activities contribute to plant resistance to abiotic stresses [42,43]. The activities of SOD and POD have been found to be higher in many species after exposure to drought stress [44,45]. Overexpression of the grapevine transcription factor *VvABF2* in *Arabidopsis thaliana* led to considerably higher SOD and POD activities in transgenic plants than in wild-type plants under drought treatment and enhanced the drought tolerance of *Arabidopsis* [46]. The bZIP transcription factor *PtrABF*, isolated from *Poncirus trifoliata*, also increased the SOD and POD activities of transgenic tobacco lines compared with those of wild type after drought treatment [47]. In our study, the POD and SOD activities in the leaves of *CmbZIP9* transgenic tobacco were significantly increased under drought stress and these plants showed more stress resistance than wild-type tobacco (Figure 5). In addition, the transcripts of the *SOD* gene and some stress-related genes (*AREB*, *SAMDC, ERF*, and *ADC*) significantly increased in the transgenic lines under drought stress in this study. Previous studies have reported that these genes are involved in the response to abiotic stress and show higher expression levels under environmental stress [48,49,50,51,52,53,54]. Taking these results together, we found that the overexpression of *CmbZIP9* led to increases in POD and SOD activities and in the expression levels of genes related to environment stress under drought stress, which might enhance the drought tolerance of plants. These results demonstrated that *CmbZIP9* plays important roles in drought stress resistance.

## 4. Materials and Methods

### 4.1. Plant Materials

The *Chrysanthemum mongolicum* plants used in this study were collected from Baotou, Inner Mongolia, China, and transplanted into the nursery garden of Northeastern University, China. A voucher specimen was collected and stored in the herbarium of Northeastern University, with voucher number ZL-20170817-02. Plant leaves were collected, immediately frozen in liquid nitrogen, and stored at −80 °C. Total RNA was extracted using a Plant RNA kit (Omega Bio-Tek, Norcross, GA, USA) and treated with DNase I (Omega Bio-Tek, USA) to remove DNA contamination. Genomic DNA was extracted from fresh leaves using a DNA kit (DNeasy Plant Mini Kit, (Qiagen, Santa Clarita, CA, USA) and treated with RNase to remove the RNA. 

### 4.2. CmbZIP Genes Identification, Motif Composition Analyses 

The bZIP family members were detected from the transcriptome of *C. mongolicum* [26] using a hidden Markov model (HMM) (http://pfam.xfam.org/, accessed on 23 October 2023). The open reading frames (ORFs) were confirmed using NCBI’s ORFfinder (ORFfinder Archives—NCBI Insights). The number of amino acids, isoelectric point (pI), and molecular weight (Mw) of the completely coded CmbZIP members were determined using the ExPASy website (https://www.expasy.org/, accessed on 2 November 2023) [55]. Thw WoLF PSORT website (https://wolfpsort.hgc.jp/, accessed on 10 November 2023) [56] was used to predict the localization of CmbZIPs. Online MEME (v 5.0) software was used to confirm the conserved motifs of the bZIPs (http://meme-suite.org/tools/meme, accessed on 15 November 2023) with a motif threshold of 10 for each sequence.

### 4.3. Phylogenetic Analysis

Seventy-one bZIP proteins of Arabidopsis thaliana were obtained from TAIR (https://www.arabidopsis.org/index.jsp, accessed on 1 December 2023). Eleven chrysanthemum bZIP protein sequences were obtained from NCBI (https://www.ncbi.nlm.nih.gov/, accessed on 5 December 2023). The putative bZIP protein sequences of *C. mongolicum* and AtbZIPs were aligned using MEGA [57] and manually checked. IQ-TREE was employed to construct a maximum likelihood (ML) phylogenetic tree with 1000 bootstrap replicates [58]. TBtools was used to visualize the phylogenetic tree and motif compositions [59].

### 4.4. Isolation of CmbZIP9 and Sequence Analysis

The quality of the total RNA was measured according to the method described by Zhang et al. [60]. First-strand cDNA was synthesized from a library of leaves using a Revert Aid First Strand cDNA Synthesis kit (Thermo Fisher Scientific Inc., Waltham, MA, USA), according to the manufacturer’s instructions. The complete *CmbZIP9* was amplified via RT-PCR and genomic PCR with primers M551ZIP F and M551ZIP R (Appendix A) which were designed according to cluster-19700.13551 in the transcriptome data of *C. mongolicum* [18] with Pfu DNA polymerase (Takara Tokyo, Japan). Purified PCR products were linked to the Kan-EZ-TA-TOPO vector and transformed into *Escherichia coli* competent cells according to the method previously described by Hu et al. [61]. Transformed colonies were identified by PCR with gene-specific primers; restriction digestion and sequenced by Zhongmeitaihe Company (Beijing, China). DNAMAN software (version 6.0) was used to align multiple amino acid sequences, and a neighbor-joining phylogenetic tree was constructed using MEGA 6 software with 1000 bootstrap replicates using Kimura two-parameter distances and the pairwise deletion of gaps. The intron–exon structures of *CmbZIP9* were analyzed using TBtools [59] and checked manually. 

### 4.5. Transformation of CmbZIP9 in Tobacco 

The coding sequence of *CmbZIP9* was inserted into the pBI121 vector to generate the 35S:: CmbZIP9 expression vector, which was transformed into *Nicotiana tabacum* using the leaf disk method, mediated by *Agrobacterium tumefaciens* strain EHA105 following the method described by Hu et al. [61]. A series of Murashige and Skoog (MS) media with kanamycin and rifampicin were used to culture the leaf disks. Plant transformation was confirmed using genomic PCR, with specific primers used for initial *CmbZIP9* amplification. The qPCR was performed on ABI QuantStudio 5 (Applied Biosystems, Foster City, CA, USA) using an SYBR Green PCR master mix kit (Thermol Fisher Scientific, Waltham, MA, USA) following the manufacturer’s instructions to explore the transcript levels of the transgenic tobacco lines. *NtACTIN* gene was used as an endogenous control [62]. 

### 4.6. Drought Stress Treatment 

Tobacco seedlings (4–5 weeks old) of the transgenic and wild-type plants were cultured as described by Li et al. [62]. Plants with 8–10 leaves were used for drought stress treatment. After complete watering, the plants were cultured at 25 °C day/18 °C night with 70% relative humidity and were not watered again. A minimum of three replicates were performed for each experiment. The leaves of treated and control plants were harvested after drought stress treatment for 0, 4, and 7 days, immediately frozen in liquid nitrogen, and stored at −80 °C until use. 

### 4.7. Measurement of Relative Water Content (RWC) 

For relative water content (RWC) measurement, we followed the method described by Yaghoubian et al. [63]. with minor modifications to the method. Briefly, the fresh weight of each leaf was determined, then the leaf was immersed immediately in distilled water at 4 °C for 24 h to obtain the saturation weight (SW) and then placed in an oven at 80 °C for 24 h to measure the dry weight (DW). 

### 4.8. Determination of MDA Content 

MDA is a biomarker of lipid oxidation in plant membrane systems and indirectly reflects the level of damage to the plant membrane system and stress tolerance. The MDA content was measured using the method described by Li et al. [62] with minor modifications. Briefly, approximately 0.1 g of leaf tissue was homogenized with 5 mL of trichloroacetic acid (5%, *w*/*v*) and then centrifuged at 3000 rpm for 10 min. Next, 0.067% (*w*/*v*) thiobarbituric acid (2 mL) was added to 2 mL of the supernatant to be mixed, boiled for 30 min, and cooled on ice immediately. Then, the solution was centrifuged at 3000 rpm for 10 min. The absorbance of the supernatant was measured at 450, 532, and 600 nm to calculate the MDA content. 

### 4.9. Measurement of the SOD and POD Activities 

POD and SOD are antioxidant enzymes that play crucial roles in plant responses to various abiotic stresses. SOD and POD activities were determined using an ELISA reader (BioTek Instruments, Winooski, VT, USA), according to the manufacturer’s instructions. About 0.3 g of leaves was homogenized with 5 mL of 0.1 mol/L Tris-HCl buffer (pH 8.5) to extract the crude enzyme solution after centrifuging at 4000 rpm for 5 min. The supernatant was used to measure the SOD content at 550 nm, and POD activity was assessed following the method described by Pan et al. [64]. 

### 4.10. Quantification Analysis of Stress-Related Genes 

The transcript levels of antioxidative gene *NtSOD* and stress-related genes, including *NtERF*, *NtSAMD*, *AREB,* and *NtADC,* were examined in the wild-type and transgenic plants under drought stress conditions. The NtActin gene was used as a reference. The qRT-PCR reaction was performed in 20 μL volumes containing 10 μL of SYBR Green Master Mix (Thermo Fisher Scientific, USA), 1.0 µL of both forward and reverse primers to a final concentration of 200 µM, 2 μL of a template diluted 1: 5 of the cDNA, and 2.5 μL of PCR-grade ddH_2_O to a final volume of 20 μL. The qPCR was performed on an ABI QuantStudio 5 real-time PCR system (Applied Biosystems, Foster City, CA, USA) with the following procedure: 55 °C for 2 min, 94°C for 5 min, and 40 cycles consisting of denaturation 94 °C for 15 s, 60 °C for 1 min. Each experiment was independently repeated in triplicate. Relative transcript levels were calculated using the 2^−ΔΔCt^ method [65]. The primers used for qRT-PCR are listed in Appendix A.

### 4.11. Statistical Analyses

Three independent samples were used for each analysis. Statistical analyses were performed using GraphPad Prism 8.0 (San Diego, CA, USA). Multiple range tests were used to detect significant differences between the mean values, and statistical significance was defined as *p* < 0.05, *p* < 0.01, or *p* < 0.001.

## 5. Conclusions

In this study, 28 CmbZIP family genes were identified in the transcriptome of *C. mongolicum,* and 10 motifs were detected among these CmbZIP proteins. A candidate gene, *CmbZIP9,* was isolated. *CmbZIP9* contains five exons and four introns. The overexpression of *CmbZIP9* increased the drought tolerance of transgenic tobacco lines. The functions of the other *CmbZIPs* we identified in *C. mongolicum* and the additional functions of *CmbZIP9* need to be further explored. 

## Figures and Tables

**Figure 1 plants-13-02064-f001:**
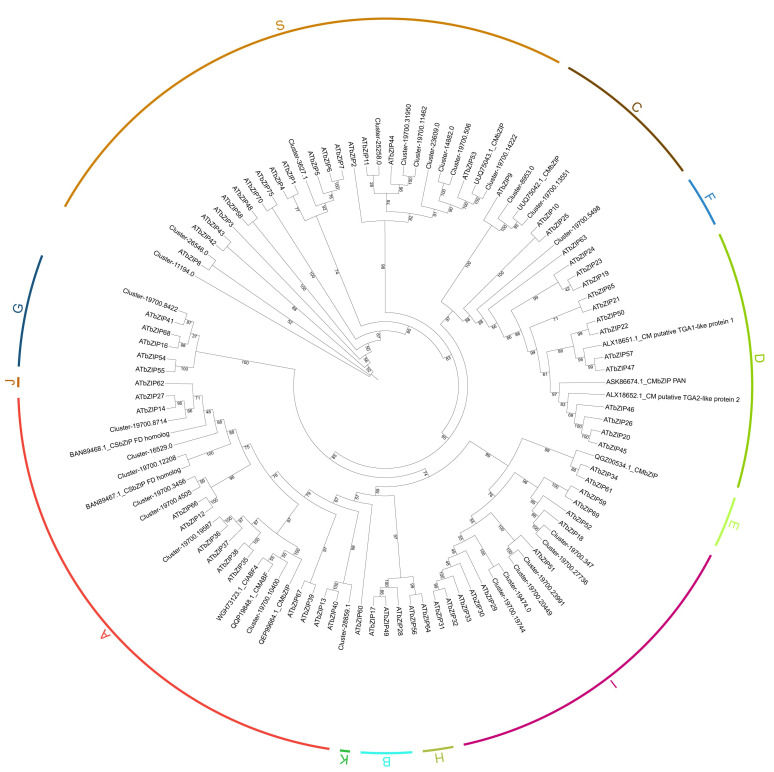
Phylogenetic analysis of bZIP proteins in *C. mongolicum*. The CmbZIPs are shown as clusters in the leaf transcriptome of *C. mongolicum*. Branch support was assessed with 1000 bootstrap replicates. The support values are provided at each node. The letters with different colors outside the circle indicate the various subgroups. AtbZIPs represent the bZIPs from *A. thaliana*. The 11 *Chrysanthemum* bZIPs are displayed with their sequence IDs and names.

**Figure 2 plants-13-02064-f002:**
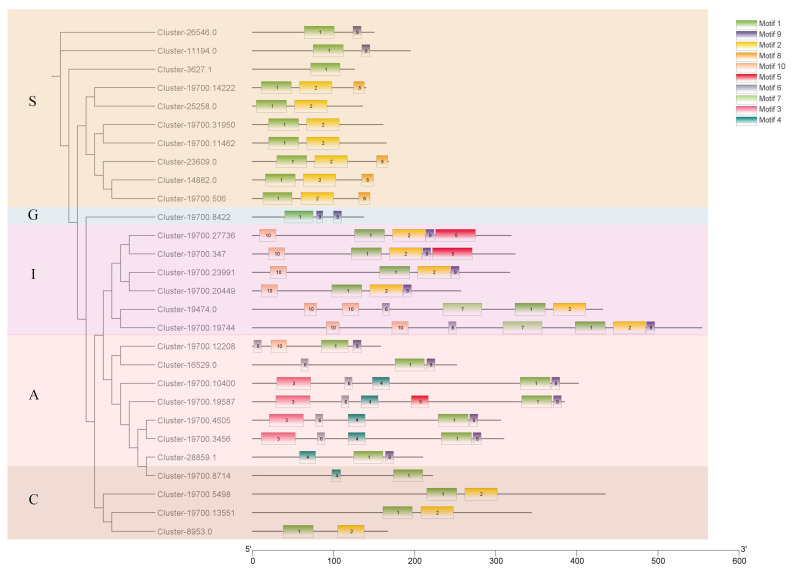
Conserved motif analysis of bZIP family in *C. mongolicum*. Conservative motifs are indicated with different colored boxes. The black line represents nonconserved sequences.

**Figure 3 plants-13-02064-f003:**
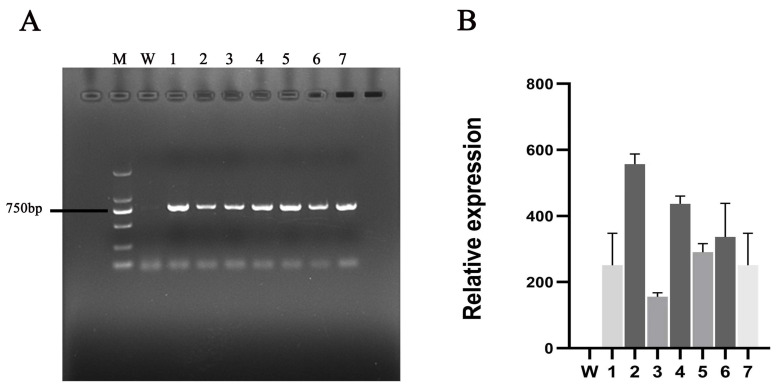
Identification of transgenic tobacco plants. (**A**) Genomic PCR for the identification of the transgenic plants. M: DL2000 marker, W: wild-type tobacco, lines 1–7: transgenic tobacco lines; uncropped gels are shown in Appendix A. (**B**) Transcript levels of *CmbZIP9* in wild-type and transgenic tobacco plants. Data are shown as mean ± SD (n = 3).

**Figure 4 plants-13-02064-f004:**
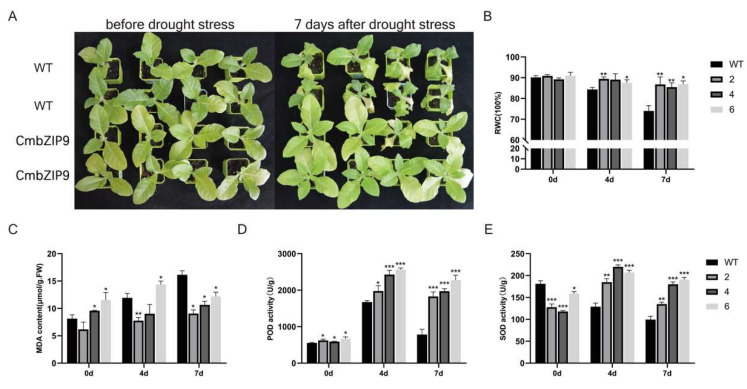
The constitutive expression of *CmbZIP9* in tobacco enhanced drought tolerance. (**A**) Morphology of wild-type and CmbZIP9 transgenic plants before and after 7 days of drought stress treatment. (**B**) Relative water content (RWC) before and after drought treatment for 4 and 7 d. (**C**) MDA content before and after drought treatment for 4 and 7 d. (**D**) Peroxidase (POD) activity before and after drought treatment for 4 and 7 d. (**E**) Superoxide dismutase (SOD) activity before and after drought treatment for 4 and 7 d. The numbers 2, 4, and 6 indicate T1 generations of three different transgenic lines. Asterisks represent values that are significantly different from those of the wild type (* *p* < 0.05, ** *p* < 0.01, *** *p* < 0.001).

**Figure 5 plants-13-02064-f005:**
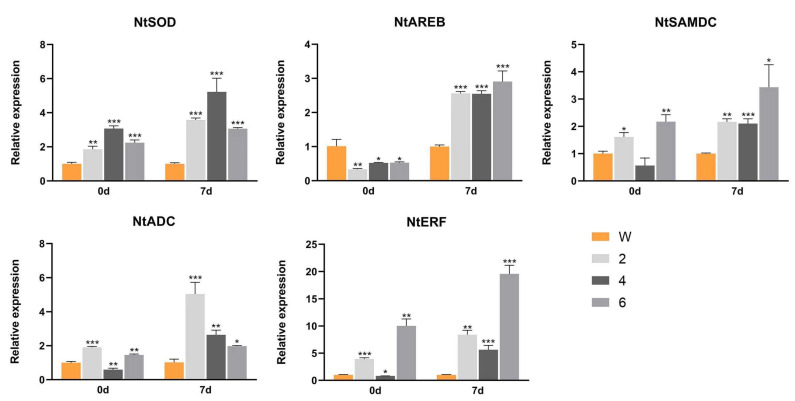
Expression levels of stress-related genes in wild-type and transgenic tobacco lines before and after 7 days of drought treatment. W: wild-type tobacco; 2, 4, and 6: T1 generations in three different transgenic lines. Data are presented as mean ± SD of three independent experiments. Asterisks represent values that are significantly different from those of the wild type (* *p* < 0.05, ** *p* < 0.01, *** *p* < 0.001).

## Data Availability

The data that support the findings of this study are openly available from NCBI GenBank at https://www.ncbi.nlm.nih.gov/. The accession numbers can be found in the article.

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
