# Peer review of "Characterization of bZIP Transcription Factors in Transcriptome of Chrysanthemum mongolicum and Roles of CmbZIP9 in Drought Stress Resistance"

_plants, 2024, doi:10.3390/plants13152064_

Round 1

Reviewer 1 Report

Comments and Suggestions for Authors

The study aims to identify and characterize the bZIP transcription factors from the transcriptome of Chrysanthemum mongolicum, a drought-tolerant plant. The research focuses on the novel bZIP gene, CmbZIP9, and its role in enhancing drought resistance when overexpressed in tobacco. The abstract provides a clear and concise summary of the study, including the identification of bZIP TFs, characterization of CmbZIP9, and its role in enhancing drought tolerance in transgenic tobacco. The key results and implications are well summarized. The introduction provides a comprehensive background on the significance of bZIP transcription factors and the challenges posed by drought stress in plants. The importance of Chrysanthemum mongolicum as a drought-resistant species is well-established. The introduction could be more succinct, with a clearer statement of the research gap and objectives.

More specific comments:

Figure 1: ATbZIPs, ‘At bZIPs’ should be ‘AtbZIPs’.

Line 75: ‘AhbZIP.’?

Line 72: ‘A. thaliana

Table S1: Physicochemical properties of bZIP gene products…The table listed characteristics of proteins but not genes.

Figure 2: Please indicate which motifs (from 1-10) are the conserved basic region and leucine zipper motif. And Table S2: The motifs characterized in CmbZIPs. ‘Sequence logos of the CmbZIP motifs’

Figure 3: Move this figure to supplementary file. Provide details on how the exons and introns were identified.

Figure 4: ‘whereas the CmbZIP-EGFP9 fluorescent signal was detected only in the nucleus (Fig. 4D–F).’ The microscopy images presented are of low quality, making it difficult to clarify the subcellular localization. At least cellular markers should be included. Please also indicate the localization signals with arrows.

Figure 5: ‘Asterisks indicate values that are significantly different from those of the wild-type plants (*p < 0.05).’ Since no expression was detected, at least not shown in Figure 5B (W: wild-type tobacco), please explain how the statistical analysis was performed. Please indicate the band size in the lane of M.

Figure 6: It is claimed that three independent transgenic lines (#2, 4, and 6) were selected for functional studies, therefore how many lines were subjected to drought treatment and morphological observation in Figure 6A.

Figure 7: It is strange that it seems the expression level of CmbZIP9 in transgenic tobacco, which is under the control of constitutive promoter, is inducible by drought stress.

Comments on the Quality of English Language

Moderate editing of English language required.

Reviewer 2 Report

Comments and Suggestions for Authors

The paper reports the identification of bZIPs in the leaf transcriptome of Chrysanthemum mongolicum, their comparative analysis, and the generation and characterization of bZIP9 transgenic tobacco plants. In my opinion, the paper can be published in Plants. However, I have concerns that should be addressed before the paper is accepted for publication.

Major points.

1. In the leaf transcriptome of Chrysanthemum mongolicum, the authors found 28 bZIPs that fall into 6 of 13 classes of Arabidopsis bZIPs, leaving open the question whether other 7 classes of Arabidopsis bZIPs have their counterparts in C. mongolicum. Therefore, it is necessary to compare the reported data on bZIPs in the leaf transcriptome of C. mongolicum with data on bZIPs in genomic sequences available for other Chrysanthemum species. 

2. Figure 4 D-F. The GFP signal is very weak. Better quality images are needed. Most importantly, these images do not indicate that CmbZIP9-GFP is localized to the nucleus, as no nuclear marker was used. Therefore, the authors' statement that CmbZIP9-GFP localizes to the nucleus is not supported by experimental data. I suggest that the section on CmbZIP9-GFP localization be removed from the paper. Alternatively, the authors can provide higher quality images showing colocalization of CmbZIP9-GFP with a nuclear marker.

Minor comments. 

The abstract is chaotic. It starts with identification of bZIPs in the C. mongolicum transcriptome, then jumps to transgenic tobacco plants, then jumps back to bZIPs found in the transcriptome, then jumps back to tobacco. Should be reworked.

Line 20. What are POD and SOD? Should be given in full in the abstract.

Lines 28-37. This introductory paragraph gives general information from textbooks and is not needed. Should be deleted.

Line 40. ABA is not decoded. 

Lines 57-58. "Excellent" is repeated twice in one sentence.

Lane 70. “Most of CmbZIP localized in nuclear.” (1) Grammar. (2) How can the authors know the localization of all these proteins? If this is a prediction, it should be clearly stated both in the text and in Table S1.

Lanes 72-72. “There were no homologs…” Such homologous proteins were not found in the obtained transcriptome data. It should be noted that these proteins may be expressed in different plant organs and/or at different developmental stages. Again, information about bZIPs in the genomes of other Chrysanthemum species is needed.

Figure 2. The drawing overlaps the legend on the left.

Comments on the Quality of English Language

Minor editing of English language required

Round 2

Reviewer 1 Report

Comments and Suggestions for Authors

the authors have made considerable improvments to the manuscript that it is acceptable for publication in the international journal PLANTS.

Comments on the Quality of English Language

Minor editing of English language required.